# High-Throughput Evaluation of Mechanical Exfoliation Using Optical Classification of Two-Dimensional Materials

**DOI:** 10.3390/mi16101084

**Published:** 2025-09-25

**Authors:** Anthony Gasbarro, Yong-Sung D. Masuda, Victor M. Lubecke

**Affiliations:** 1Graphene Microfluidic Laboratory, Naval Information Warfare Center Pacific, Pearl City, HI 96782, USA; 2Department of Electrical Engineering, University of Hawai‘i at Mānoa, Honolulu, HI 96822, USA; ydmasuda@hawaii.edu (Y.-S.D.M.); lubecke@hawaii.edu (V.M.L.)

**Keywords:** 2D materials, graphene, machine learning, clustering, image processing, optical classification, unsupervised learning, parallel computing

## Abstract

Mechanical exfoliation remains the most common method for producing high-quality two-dimensional (2D) materials, but its inherently low yield requires screening large numbers of samples to identify usable flakes. Efficient optimization of the exfoliation process demands scalable methods to analyze deposited material across extensive datasets. While machine learning clustering techniques have demonstrated ~95% accuracy in classifying 2D material thicknesses from optical microscopy images, current tools are limited by slow processing speeds and heavy reliance on manual user input. This work presents an open-source, GPU-accelerated software platform that builds upon existing classification methods to enable high-throughput analysis of 2D material samples. By leveraging parallel computation, optimizing core algorithms, and automating preprocessing steps, the software can quantify flake coverage and thickness across uncompressed optical images at scale. Benchmark comparisons show that this implementation processes over 200× more pixel data with a 60× reduction in processing time relative to the original software. Specifically, a full dataset of2916 uncompressed images can be classified in 35 min, compared to an estimated 32 h required by the baseline method using compressed images. This platform enables rapid evaluation of exfoliation results across multiple trials, providing a practical tool for optimizing deposition techniques and improving the yield of high-quality 2D materials.

## 1. Introduction

Two-dimensional (2D) materials have attracted significant interest in recent years due to their unique physical properties and potential applications in electronics, including superconductors for quantum computing and high-precision RF sensors [1,2,3,4]. Among these, graphene remains the most prominent example [5]. Although other methods such as chemical vapor deposition, liquid exfoliation, and electrochemical exfoliation exist, mechanical exfoliation of bulk crystals continues to be the most widely used technique for producing high-quality, single-grain flakes. However, this process is labor-intensive and yields a low number of usable flakes, necessitating the preparation of large numbers of samples to identify candidates [6,7,8]. This publication focuses on quantifying the yield of mechanically exfoliated 2D materials using optical classification techniques. The method presented here can be universally applied to any mechanically exfoliated 2D material that can be deposited on a substrate and then identified through optical contrast, including graphene, MoS_2_, MoSe_2_, WS_2_, WSe_2_, hBN, and others.

Mechanical exfoliation parameters, such as tape removal speed and angle, have been shown in theoretical studies to significantly affect flake deposition, yet they are often neglected in practice in favor of improving yields through generating high volumes of samples [9]. Most researchers still rely on manual inspection through optical microscopes to identify suitable flakes, making it difficult to systematically evaluate the impact of exfoliation conditions. As a result, little practical work has focused on optimizing the process for higher-yield production of 2D materials. To enable such optimization, robust software tools are needed to quantify flake coverage and thickness across large, diverse sample sets.

Several recent studies have proposed automated optical classification techniques for 2D materials. However, these are typically designed only to assist in identifying flakes for device fabrication, not for large-scale statistical analysis. For example, some tools require manual region selection in microscopy images [10], while others use supervised learning methods that demand extensive labeled datasets [11]. Another approach provides real-time feedback during manual scanning, but is not suited for batch processing of large datasets [12]. Limitations, such as dependence on user input and lack of scalability, have hindered their use in process optimization studies.

This work builds on a previously published platform designed for rigorously controlling and quantifying the tape removal angle and speed in order to reliably reproduce mechanical exfoliation parameters [13]. Presented here is an improved, open-source software tool that integrates GPU acceleration, automated image preprocessing, and parallelized execution for high-throughput optical classification of 2D materials. The software extends an existing state-of-the-art unsupervised clustering method, improving upon its execution time limitations to allow for rapid analysis of large datasets with minimal user intervention [10]. It combines traditional image processing techniques with machine learning to classify flake thicknesses based on optical contrast. An overview of the software workflow is shown in Figure 1.

## 2. Materials and Methods

### 2.1. Background

Mechanical exfoliation of 2D materials involves repeatedly peeling layers from a bulk crystal using adhesive tape to thin the flakes attached. These flakes are then transferred to a target substrate, such as SiO_2_, where they are typically identified by visually inspecting optical contrast under a microscope. While effective for isolated sample preparation, this manual inspection process is impractical for large-volume applications requiring statistical evaluation of numerous samples.

The inability to efficiently process large datasets makes it difficult to assess how exfoliation parameters affect flake yield and quality. To address this challenge, we develop an open-source software platform that automates the classification of 2D material flakes using image processing and machine learning methods. Building on recent developments in the field [10,11,12], this tool enables rapid statistical analysis of deposited flake area and thickness across a wide range of samples, providing a foundation for optimizing exfoliation procedures.

### 2.2. Software Overview

The software platform was developed by adapting an existing open-source segmentation model previously shown to achieve high classification accuracy for 2D materials [10]. While achieving desirable results, the original implementation was not optimized for large-scale datasets, creating problems when trying to analyze trends across many samples. Training the previous version of the model on just 10 images reportedly required 10 h and images classification was estimated to take approximately one minute to process each image. For a practical example, scanning a 15 × 20 mm substrate using a 20× objective at 2048 × 1536 resolution yields 2916 images and processing would take over 48 h at this rate. Furthermore, the model required manual user input to define the substrate background for each image, which is impractical when processing thousands of images across multiple samples.

To address these limitations, the software must be adapted to meet three key requirements: (1) minimize manual user input, (2) significantly accelerate the classification process while maintaining accuracy, and (3) enable automatic export of layer thickness and area data for high-volume statistical analysis. The Python version 3.11.9 libraries used are listed in Section A.1.

The platform operates in two phases: a training phase and a testing phase. During training, users provide cropped images of individual flakes representing known thicknesses. These examples are preprocessed and clustered in RGB color space to identify distinct optical layer regions. The resulting clusters are then labeled by thickness (in layer count), and this data is stored in a master catalog. Each master catalog is trained on a specific set of conditions (e.g., material type, substrate, lighting), but can be reused to classify other images captured under relatively similar conditions. The system is designed to easily be adapted to new materials and substrates by quickly retraining another master catalog for each use case. Retraining takes approximately 45 s per image, and is effective at classification with as few as 1 or 2 examples of each classification category, allowing users to rapidly build catalogs for different conditions and materials.

In the testing phase, full-size images are processed by comparing their pixel values to the master catalog. Each pixel is classified by thickness group, and area statistics are computed for each category. The software is designed for use on systems with Graphics Processing Units (GPUs), enabling parallelized execution that dramatically improves processing speed over the original CPU-bound implementation.

#### 2.2.1. Image Processing Pipeline

In the training phase, users manually select flake-containing regions and define a rectangular crop for each image through the software interface. A small area of background is also selected and masked to enable normalization during preprocessing. The preprocessing pipeline includes bilateral filtering for noise reduction and planar background correction based on the masked region. This image filtering and normalization process compensates for variations that may be present throughout optical scans taken from a particular microscope setup.

Each processed image is then clustered in RGB color space using mean-shift clustering. The resulting pixel groups are refined using Density-Based Spatial Clustering of Applications with Noise (DBSCAN), followed by fitting to Gaussian Mixture Models via Expectation Maximization (GMM-EM) to determine the ellipsoidal parameters that best describe the data distributions. These cluster descriptors are stored in a catalog file, which serves as the basis for classifying full-resolution images during the testing phase.

This cataloging pipeline prioritizes classification accuracy over raw speed. In contrast to related works that employ K-means clustering, this approach uses mean-shift clustering for its adaptability. While K-means is a faster, centroid-based algorithm with linear time complexity, it requires the number of clusters to be specified in advance and performs poorly when clusters vary in shape and size. Mean shift, a density-based method, avoids this limitation by discovering cluster structures directly from the data without prior assumptions [14,15].

#### 2.2.2. GPU Acceleration

The original segmentation software was introduced as a proof of concept for unsupervised clustering, and was not optimized for performance. Its image processing pipeline relied heavily on nested Python loops executed sequentially on a single CPU thread, resulting in significant execution bottlenecks when processing large image sets.

To address these limitations, enhancements were made of the existing open-source implementation to support GPU acceleration using CuPy version 13.5.1, a Python library for numerical computation built on Nvidia’s CUDA (Compute Unified Device Architecture) platform. CuPy is designed as a near drop-in replacement for NumPy and SciPy, enabling GPU-based execution with minimal code modification [16,17].

In traditional CPU-based implementations, operations such as pixel-wise transformations (e.g., brightening an image) require iterating through each pixel in sequence, consuming many instruction cycles. In contrast, CuPy translates such operations into precompiled CUDA kernels that launch a grid of parallel threads, each responsible for a single pixel. This massively parallel execution enables all pixels to be processed simultaneously in a single operation.

All image processing components in the pipeline were reimplemented using CuPy to achieve full parallelization while maintaining functional equivalence with the original code. Hardware acceleration of image processing has been widely adopted in research due to its ability to dramatically reduce runtime for computationally intensive tasks [18,19]. A summary of the resulting gains in time complexity is presented in Table 1.

#### 2.2.3. Automatic Background Masking

The classification pipeline was redesigned for full automation to eliminate requirements for manual user intervention. Mechanical exfoliation procedures can be followed by automated scanning of entire sample surfaces using optical microscopy, producing large directories of high-resolution images. Once a master catalog has been trained, the software can be applied directly to these image directories, performing batch classification and exporting results for downstream analysis without user input.

Originally, the preprocessing stage required users to manually define a background region for each image to support planar fit normalization. This process was replaced with an automated background detection method based on GPU-accelerated local variance thresholding, which identifies substrate regions based on their uniform texture [20,21].

The algorithm begins by applying bilateral filtering to reduce noise. It then calculates local variance for each pixel using a sliding window, evaluating both grayscale and RGB channels. A combined variance image is created, and then a binary background mask is generated by classifying pixels below the variance threshold as background. Morphological dilation and erosion operations are applied to refine the mask by removing artifacts and filling small gaps.

Automating background mask generation enables complete image classification to proceed without any required user annotations. An example of a generated background mask and the corresponding preprocessed image is shown in Figure 2.

#### 2.2.4. Data Export and Statistical Analysis

Testing images were captured using a 25 × 15 mm SiO_2_ substrate on an HQGraphene HQ2D MOT 2D (HQGraphene, Groningen, The Netherlands) material transfer station. The entire surface was imaged using a 20× objective lens with automated scanning, resulting in a dataset of 2916 images at 2048 × 1536 resolution.

During classification, visualizations of each processing step comparing the original image with layer-classified regions can be automatically exported as PNG files. These overlays provide user feedback and display each flake region color-coded by thickness labels. This visualization feature can be optionally disabled to reduce processing time when only data output is required.

In addition to image overlays, a CSV file is generated for each classified image. This file contains structured data for postprocessing and statistical analysis. Each row represents a single flake region and includes the following fields: image filename, region ID, estimated thickness (in number of layers), area (in pixels), and mean RGB values of the region. This standardized output format facilitates downstream analysis using tools such as Python, R, or spreadsheet software.

## 3. Results

### 3.1. Performance Benchmarking

Performance benchmarking was conducted on a ThinkPad P1 Gen 6 (Lenovo, Beijing, China) laptop equipped with an Nvidia RTX 4090 Max-Q GPU, an Intel i9-13900H CPU, and 64 GB of RAM. All tests compared the original single-threaded software against the GPU-enhanced implementation. Notably, the original software applied image compression during processing, reducing the number of pixels analyzed and offering a computational advantage. In contrast, the vectorized version processed full-resolution, uncompressed images. A summary of the benchmark results is shown in Figure 3.

Training performance was evaluated using a cropped flake image from the original dataset. The original software, using a compressed 90 × 112 image (10,080 pixels), required 2835.77 s (~47 min) to complete training. In comparison, the GPU-accelerated implementation was trained on the full 800 × 1000 uncompressed image (800,000 pixels) in only 44.52 s—representing a 63× increase in speed, despite operating on nearly 79× more pixel data.

Classification performance was benchmarked using the same full-size 4912 × 3684 image (18,095,808 pixels). The GPU implementation completed classification in 0.67 s, while the original software, using a compressed 347 × 260 version (90,220 pixels), required 40.44 s. This equates to a 60× reduction in processing time while processing over 200× more data.

In large-scale testing, the GPU-accelerated software can classify an entire uncompressed dataset of 2916 images in approximately 35 min. The original software, by extrapolation, would require over 32 h to process the same dataset. These performance gains are especially notable given the additional required computation overhead in the GPU version for automatic background masking and processing uncompressed pixel data.

### 3.2. Classification Accuracy

Classification accuracy was validated using the test images and ground truth data from the original publication [10]. The GPU-accelerated software achieved an average pixel-wise classification accuracy of approximately 95%, consistent with the original implementation. Ground truth masks were manually recolored and used to generate confusion matrices and error visualizations, as shown in Figure 4.

Additional classification examples are provided in Section A.2.

## 4. Discussion

This work presents an open-source software platform designed to automate the classification of large datasets of optical microscopy images for evaluating 2D material deposition yields. The platform is lightweight enough to run on consumer-grade laptops equipped with dedicated GPUs. By leveraging GPU acceleration, the software achieves significant speed improvements while maintaining high classification accuracy. The ability to process uncompressed images allows for preservation of subtle optical contrast features that might otherwise be lost during preprocessing or compression.

Reducing the training time from approximately 47 min to 44 s enables rapid iteration when building or refining the classification catalog. This transforms catalog generation from an all-day task to one that can be completed in under an hour, significantly lowering the labor cost associated with dataset preparation for different materials or substrates.

The platform is designed for minimal user input. A key feature is the automated background masking algorithm, which eliminates the need for manual cropping by identifying background regions based on local texture variance. This capability enables fully automated classification of large directories of high-resolution images, supporting scalable analysis of exfoliation procedures across many test samples.

### Limitations

An individual classification master catalog is required to be trained for each specific combination of 2D material, substrate, and microscope scanning setup. This software does not currently support transfer learning or domain adaptation techniques that could allow a single catalog to generalize across multiple scenarios. This requirement is minimized by the speed at which a software catalog can be trained and the low amount of image data necessary. Catalog training requires less than a minute per image and can effectively classify with as few as one or two examples from each cluster category. The classification accuracy testing detailed in Section A.2 was achieved using a single training flake image to generate each catalog for a material-on-substrate combination. Training and testing images are available from the original publication [10] and the software source code’s GitHub Page: https://github.com/AQUAMAG/hardware-accelerated-2d-material-classifier, accessed on 15 August 2025.

The current implementation relies on the CuPy Python library, and is therefore limited to systems with Nvidia GPUs with CUDA capability. The codebase can be adapted to run on CPU-only systems with minor modifications to replace CuPy with NumPy, albeit with greatly reduced performance. While it is technically possible for it to run on CPU-only systems, at least a single dedicated GPU is typically considered to be required for applications of image processing in general. All benchmarks were performed on a laptop with a single mobile GPU, demonstrating that high-end desktop hardware is not required. This is not a significant limitation, as consumer-grade laptops with dedicated GPUs are widely available and affordable. Lower-specification GPUs may result in longer processing times, but are expected to perform adequately for most use cases.

Automatic background masking assumes a relatively uniform and consistent substrate texture. In cases where the background is highly variable or non-uniform, the automated method may fail to accurately identify the substrate. In such instances, users may revert to manual background masking to ensure correct classification. The software testing function retains the option for manual definition of a background region if needed, and the existing catalog can be rerun on these images separately.

The flake identification algorithm performs best with monolayer and few-layer flakes. Thicker flakes, which can sometimes overlap with residues in RGB space, could have small portions that may be misclassified. This was greatly mitigated by treating thicker flakes as a single classification group representing a range of layers, along with training to classify residue as a different category. As a result of these training changes, the software can distinguish between residue and thick flakes as whole classification categories. This does not pose a significant concern, because thicker flakes are typically of less interest for 2D material applications and are not expected to require individual layer thickness analysis. The software is optimized to classify the bulk area of flakes, enabling meaningful statistical comparisons of deposition results across samples.

## 5. Conclusions

In summary, this publication presents a GPU-accelerated, open-source software platform for the rapid and automated classification of 2D material thicknesses from optical microscopy images. By extending and optimizing an existing unsupervised clustering framework, the platform achieves classification accuracy of approximately 95% while processing over 200× more pixel data in a fraction of the time. The method presented here can be universally applied to any mechanically exfoliated 2D material that can be deposited on a substrate then identified through optical contrast, including graphene, MoS_2_, MoSe_2_, WS_2_, WSe_2_, hBN, and others.

The software enables high-throughput analysis of mechanically exfoliated samples, reducing training and classification times by over 60×. This facilitates systematic studies of deposition conditions across large datasets, supporting efforts to optimize exfoliation techniques and improve the yield of high-quality 2D materials.

## Figures and Tables

**Figure 1 micromachines-16-01084-f001:**
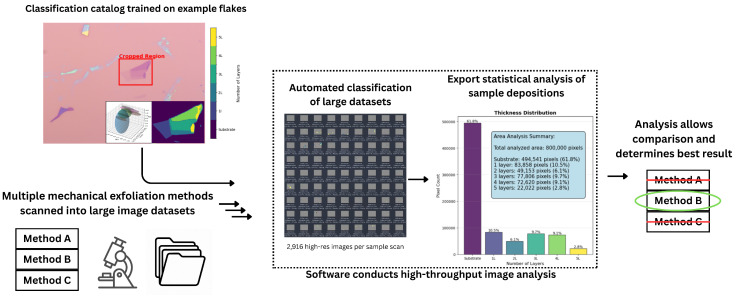
Overview of software platform. Training data is preprocessed and then clustered in RGB space using mean-shift clustering, grouped using Density-Based Spatial Clustering of Applications with Noise (DBSCAN), and fit to arbitrary normal distributions using a Gaussian Mixture Model with Expectation Maximization (GMM-EM), after which clusters are manually labeled in a catalog by material thickness. The training catalog is used to rapidly (~0.7 s per 2048 × 1536 pixel image) classify full-size images automatically for a large dataset. The final results are exported to a CSV file containing the area and thickness of each classified region, allowing for comparison to determine the efficacy of different deposition techniques.

**Figure 2 micromachines-16-01084-f002:**
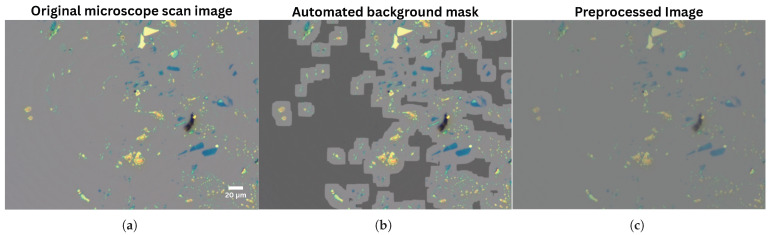
Steps in the preprocessing pipeline when using automated background masking. (**a**) Original 20× magnification full-size microscope image scan, 2048 × 1536 pixels. (**b**) Mask generated from automatic background detection, highlighted as darker gray region. (**c**) Final image after completion of preprocessing, application of a bilateral filter to reduce noise, then use of the detected background-mask as a baseline to normalize substrate RGB values via planar fit.

**Figure 3 micromachines-16-01084-f003:**
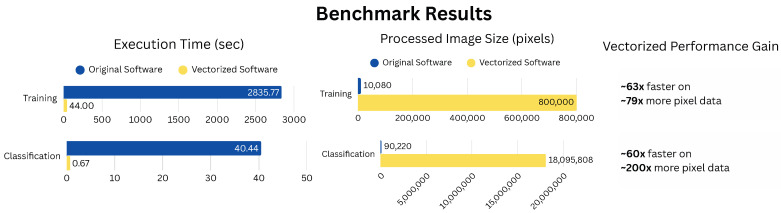
Execution time comparison showing a 63× speedup in training and 60× speedup in classification using the vectorized GPU implementation. Despite processing full uncompressed images and running additional background detection steps, the new method significantly outperforms the original, which uses image compression and predefined masks.

**Figure 4 micromachines-16-01084-f004:**
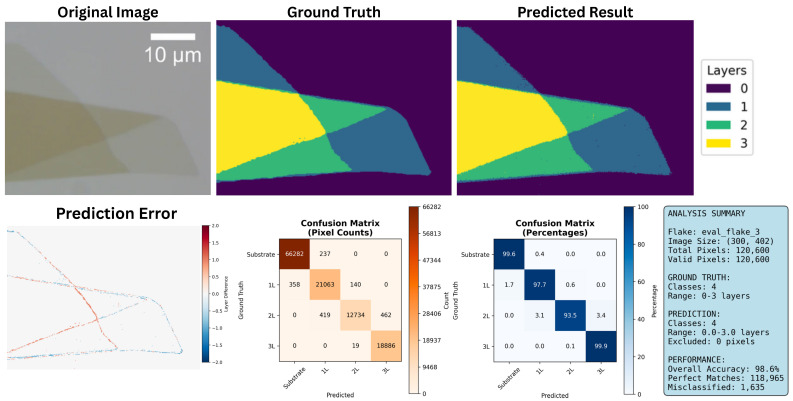
Example test from image illustrating similar classification accuracy to baseline method. Original image is of MoS_2_ on PDMS substrate used for testing [10]. Ground truth regions were manually labeled and compared to the software output by raw pixel count, resulting in an average prediction accuracy of ~95%. Confusion matrix and pixel error visualization are included for comparison.

**Table 1 micromachines-16-01084-t001:** Overview of major computation optimizations.

Operation	Optimization
Overall Execution Order	CPU sequential iterations → GPU parallelized batches
Multivariate Gaussian Computation ^1,2^	$O\left(n\cdot m^{2}\right)\to O\left(n\cdot m\right)$
Mean-Shift Distance Computation ^1,3,4^	O(n·w·h)→O(n)

^1^* n* = number of pixels; ^2^* m* = number of clusters; ^3^* w* = width of image in pixels; ^4^* h* = height of image in pixels.

## Data Availability

The software source code is available on the GitHub Page: https://github.com/AQUAMAG/hardware-accelerated-2d-material-classifier, accessed on 15 August 2025. The original contributions presented in this study are included in the article. Further inquiries can be directed to the corresponding author.

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
