# Peer review of "High-Throughput Evaluation of Mechanical Exfoliation Using Optical Classification of Two-Dimensional Materials"

_micromachines, 2025, doi:10.3390/mi16101084_

Round 1

Reviewer 1 Report

Comments and Suggestions for Authors

"This work introduces a GPU-accelerated, open-source software that significantly enhances the speed and scalability of classifying 2D material thickness from optical images, enabling high-throughput analysis to optimize exfoliation processes and offering valuable insights for the 2D materials research community. Overall, the manuscript is well written and is recommended for publication after addressing a few concerns.

  1. Author proposed a Automated Background Mask to pre-process the optical image. How to preceed with optical images taken by differnet optical image system?  
  2. Since defects are commonly present in exfoliated 2D materials, can this algorithm also be used for defect detection?
  3. Can this algorithm differentiate between tape or polymer residues and actual 2D material flakes?

Reviewer 2 Report

Comments and Suggestions for Authors

In the research paper which titled “High-Throughput Evaluation of Mechanical Exfoliation Using

Optical Classification of 2D Materials”, authors developed an open-source, GPU-accelerated software platform to automatically classify 2D material flakes by thickness from optical microscopy images. The study is interesting, however, it needs to consider the following major revision before publishing in Eng:

  • The training phase requires examples of known flake thicknesses. How easily can the catalog be transferred to different 2D materials (e.g., BP, MoS₂, hBN, WSe₂)? Does the method require retraining for each material/substrate combination?
  • In the introduction part, authors need to highlight couple of 2D materials such as BP, hBN, MoS2 and couple of exfoliation methods such as mechanical, liquid, and electrochemical exfoliation. Then, narrow down to the goal of this paper. Authors can use the following references: https://doi.org/10.1063/5.0090717, https://doi.org/10.1002/sstr.202000148.
  • The authors report ~95% accuracy, but thicker flakes and residues may overlap in RGB space. How does classification accuracy degrade as the number of layers increases, and what is the misclassification rate for multilayer flakes (>10 layers)?
  • In this study, how sensitive is the classification to variations in illumination, objective lens type, numerical aperture, or camera calibration? Would identical flakes be classified consistently across different microscope setups?
  • The paper shows a major speed improvement in catalog training, but how many flakes and thickness categories are minimally required to build a reliable catalog? How does catalog completeness affect accuracy in new datasets?
  • Full automation is emphasized, but in cases of failure (e.g., background misclassification, ambiguous flakes), does the software allow user intervention or correction without retraining the catalog?
  • The implementation relies on CuPy and Nvidia GPUs. How feasible is it for laboratories without high-end GPUs to use this software?
  • Also, a comparison of this study with the state-of-the-art research is important.

Round 2

Reviewer 2 Report

Comments and Suggestions for Authors

Ready to publish in the present form.